# G-RRDB: An Effective THz Image-Denoising Model for Moldy Wheat

**DOI:** 10.3390/foods12152819

**Published:** 2023-07-25

**Authors:** Yuying Jiang, Xinyu Chen, Hongyi Ge, Mengdie Jiang, Xixi Wen

**Affiliations:** 1Key Laboratory of Grain Information Processing & Control, Ministry of Education, Henan University of Technology, Zhengzhou 450001, China; jiangyuying11@163.com (Y.J.); xinyuchen_cc@163.com (X.C.); jj_meng3@163.com (M.J.); 15090608201@163.com (X.W.); 2Henan Provincial Key Laboratory of Grain Photoelectric Detection and Control, Zhengzhou 450001, China; 3School of Artificial Intelligence and Big Data, Henan University of Technology, Zhengzhou 450001, China; 4College of Information Science and Engineering, Henan University of Technology, Zhengzhou 450001, China

**Keywords:** moldy wheat, terahertz image, image denoising, dense residual structure

## Abstract

In order to solve the problem of large image noise and unremarkable features caused by factors such as fluctuations in the power of a light source during the terahertz image acquisition of wheat, this paper proposes a THz image-denoising model called G-RRDB. Firstly, a module called Ghost-LKA is proposed by combining a large kernel convolutional attention mechanism module with a Ghost convolutional structure, which improves the characteristics of the network to acquire a global sensory field. Secondly, by integrating a spatial attention mechanism with channel attention, an attention module called DAB is proposed to enhance the network’s attention to important features. Thirdly, the Ghost-LKA module and DAB module are combined with the baseline model, thus proposing the dense residual denoising network G-RRDB. Compared with traditional denoising networks, both the PSNR and SSIM are improved. The prediction accuracy of G-RRDB is verified through the classification of the VGG16 network, achieving a rate of 92.8%, which represents an improvement of 1.7% and 0.2% compared to the denoised images obtained from the baseline model and the combined baseline model with the DAB module, respectively. The experimental results demonstrate that G-RRDB, a THz image-denoising model based on dense residual structure for moldy wheat, exhibits excellent denoising performance.

## 1. Introduction

Wheat is the largest and most widely planted grain crop in China, ranking second only to corn in terms of production. It is known as one of the world’s top three food crops, along with corn and rice, and is an indispensable food. After harvest, wheat is susceptible to dew and mold due to factors such as storage temperature and moisture, which poses a health threat to humans and poultry. Therefore, the rapid and accurate detection of moldy wheat is the focus of this research. Currently, the main traditional detection methods include sensory evaluation (such as smell), machine vision [1] and liquid chromatography [2]. However, sensory evaluation is highly subjective, has poor repeatability and has low detection accuracy. It is challenging for machine vision to identify wheat at the early stage of mold, and liquid chromatography is time-consuming and does not meet the needs of nondestructive testing. Therefore, there is an urgent need to explore a rapid and nondestructive method for moldy wheat detection.

Terahertz (THz) [3] radiation, also known as THz waves, is electromagnetic radiation with a frequency range between 0.1 and 10 THz and a central frequency of 1 THz. It falls between the microwave and infrared bands and is located in the transition region between macroscopic electronics and microscopic photonics. Due to the presence of rotational and vibrational jumps between or within molecules, skeletal vibrations of nucleic acid macromolecules, and low-frequency vibrational absorption frequencies of lattices in crystals within the THz band, THz technology is well suited for detecting biological macromolecules. THz waves have technical characteristics such as nondestructive penetration, low photon energy, fingerprint spectrum and high bandwidth. They have been used in the safety inspection of agricultural products [4,5], biomedical [6,7], security inspection [8] and various other fields. With the continuous development of THz technology, THz imaging has gradually become an important technical tool in multidisciplinary crossover fields.

In recent years, THz waves have made significant breakthroughs in agricultural product quality inspection due to their technical advantages, such as fast imaging speed and non-destructive penetration. However, during the process of THz image acquisition, due to the interference of factors such as slight fluctuations in light source power, problems such as low image contrast and resolution and the presence of noise in the images can arise, resulting in the loss of crucial image information and edge details, consequently affecting subsequent tasks such as image classification. Therefore, the key challenge for image denoising tasks lies in filtering out noise from noisy images and restoring high-quality images. Traditional image-denoising methods primarily rely on spatial and frequency domains. Samuel et al. [9] proposed an algorithm for denoising THz images using a nonlinear spatial function with median-mean filtering. They added Gaussian noise and pretzel noise to the images, respectively, and used a sliding window 5 × 5 and 7 × 7 joint mean-median filtering method to generate denoising results. The results demonstrated that the method has a better denoising effect on Gaussian noise. With further in-depth research, an increasing number of denoising algorithms based on convolutional neural networks have been proposed. Jiang et al. [10] developed a THz spectral image-denoising model called CBDNet-V to address the issues of poor quality and unremarkable features in original THz spectral images of imperfect wheat grains, and the denoised images obtained by this model showed improved peak signal-to-noise ratio (PSNR) and structural similarity (SSIM) compared to the denoising results of traditional models. Balaka et al. [11] applied the unsupervised learning network CycleGAN to generate pairs of noisy synthetic images produced by a handwriting generator. They trained the deep learning model Pix2pixGAN for THz image denoising with the generated noisy images. The results indicated that 99% of the characters on one side of the rice paper could be clearly recognized, while only 61% of the characters on one side of the standard paper were recognizable. This demonstrates that Pix2pixGAN has a good denoising effect.

In this paper, we utilize THz imaging technology combined with a deep learning algorithm to conduct research on the recognition of wheat mold degree. To address the issues of noise and unremarkable features in the original THz images, we propose a THz image-denoising model called G-RRDB based on deep learning algorithm. This model consists of a network of five densely connected residual modules (RRDB) [12] cascaded as the baseline model, and Ghost-LKA module and DAB module are added to the baseline model. Its denoising performance is significantly higher than that of traditional denoising networks based on deep learning. The contributions of this paper can be summarized by these three points. Firstly, combining the large kernel convolutional attention module (LKA) [13] with the architecture of Ghost convolution, we propose a large kernel attention module called Ghost-LKA based on a Ghost convolutional structure. This module enables better acquisition of coarse-grained image features from the global perceptual field. Secondly, an improved attention mechanism module named DAB is proposed, based on Channel Mechanism (CA) [14] and Spatial Attention (SA). This module adjusts the weights distribution on different features to reduce the loss of image detail information, thereby achieving effective image denoising. Thirdly, we design a dense residual-based denoising network called G-RRDB, which maximizes the acquisition of convolution information at each layer through its dense connection mechanism. This ensures that the output feature map contains more feature information from the original image.

## 2. Materials and Methods

### 2.1. Experimental Equipment and Principles

In this experiment, the THz 3D chromatography imaging system of Qingyuan Fengda was used, and its system schematic is shown in Figure 1. The detection system consists of four parts in total, namely the transmitting and receiving ends, the optical path system, the sample stage and the mobile system, with a bandwidth between 0.1 and 3.5 THz and a spectral dynamic range greater than 60 dB. In this paper, the reflection imaging mode is used for the experimental sample, so the THz transmitting and receiving antennas are distributed on the same side of the sample. Firstly, the sample is placed on the 2D scanning platform, and in the transmitting module, the photoconductive antenna converts the femtosecond pulse signal into a picosecond THz pulse signal, and a short THz pulse is emitted from the transmitting source and focused on the sample surface through the optical path system. The incident THz pulse can penetrate the sample under test because of its high penetration into nonpolar substances. Due to the different refractive indices between different thicknesses inside the sample, when the THz wave crosses the interface of different thicknesses, a part of the reflected wave is emitted, and after the reflected wave is received by the detector, the detected THz signal is converted into an electrical signal by the receiver module.

The scanning frequency of the experimental setup is set to 30 Hz, the scanning time range is set to 90 ps, the THz wave incidence rate angle is set to 25°, and both X and Y step spacings are set to 0.2 mm. During the scanning of the sample by THz instrument, the THz time-domain spectral signals from all the pixel points are assembled to form a THz image of the sample.

### 2.2. Sample Preparation and THz Image Data Acquisition of Moldy Wheat

The wheat variety used in the experiment was 22 fine wheat. This variety of wheat is characterized by drought tolerance, disease resistance, collapse resistance and high yield. Its kernels are full, hard and white, with a protein content of 12.2% and wet gluten content of 31.4%. The wheat was placed in a constant temperature and humidity chamber at 35 °C and 95% relative humidity for incubation, and beginning on the third day, the wheat gradually develops mold. The wheat was taken out at different mold growth stages on the zero, third, sixth and ninth days, respectively, and we used the samples obtained on four occasions with different levels of mold as normal wheat, slightly moldy wheat, moderately moldy wheat and seriously moldy wheat. Some samples with the four different mold levels were obtained as shown in Figure 2.

Four groups of samples were selected for the experiment: normal wheat, slightly moldy wheat, moderately moldy wheat and heavily moldy wheat. The samples were evenly placed on the 2D scanning platform and reflectance imaging was performed. The maximum scanning area of the system is 100 mm × 100 mm with a spatial resolution of 0.1 mm. A total of 9000 time domain points of THz waveforms were acquired for each pixel in a scanning time range of 90 ps. The acquired image data were stored in 3D form, containing both spatial and spectral information. Taking slightly moldy wheat as an example, the 3D data of the THz image of the 80th layer is shown in Figure 3, where different colors indicate different THz wave reflection intensities. As can be seen, the wheat kernel outline information can be observed.

### 2.3. Related Technology Principles

#### 2.3.1. Image Denoising

Image denoising refers to certain suppression and elimination of image noise to recover the true image from the image corrupted by noise [15]. Traditional image-denoising methods are divided into three categories: spatial-domain-based denoising methods, transform-domain-based denoising methods and image-compression-based denoising methods. The spatial-domain-denoising methods can be divided into neighborhood-based null filtering and non-local-based null filtering, according to the filtering range, among which, neighborhood-based null filtering includes mean filtering and median filtering, which are characterized by simple and direct, low computational complexity, but poor denoising performance; non-local-based null filtering extends the search range to a large image area, and the representative methods include Non-Local Means (NLM) etc., which is characterized by clear texture of the processed image, but relatively large computational resource consumption. The transform domain filtering technique processes the image in the frequency domain, which is based on the fact that the frequency domain noise obeys irregular distribution and is distributed across all frequency bands, but is strongest in the high frequency band. Compared with the spatial-domain processing method, the robustness of the transform-domain processing method is higher. The core idea of image-compression methods is to treat the image as a matrix, in which, the effective signal is concentrated in a limited number of image locations and the image signal intensity in the rest of the locations is weak. These methods require continuous iterations until the fit converges and the image resolution is restored to a high level, but the model is less robust. With the development of deep learning, more and more convolutional neural network (CNN)-image-denoising algorithms with good performance have emerged, such as denoising convolutional neural network (DnCNN) [16], FFDNet [17] and convolutional blind denoising network (CBDNet) [18], which have faster computation speed, better image recovery and better overall denoising performance compared with traditional methods.

#### 2.3.2. Residual Structure

Usually, the deeper the depth of the network model, the better the model performance should be, but in fact, when the network reaches a certain depth, its accuracy tends to saturate, and as the network depth continues to increase, the accuracy will show a decreasing trend. The reason for this is that the backpropagation mechanism keeps updating the parameters during the training of the deep model, but the gradient of the neural network keeps decaying during the backpropagation; thus, when the network is too deep, the gradient will disappear and the accuracy of the model will decrease. Therefore, He et al. [19] introduced the residual structure in the model to make the model more accurate at very deep depths, thus solving the above problem. The residual structure is shown in Figure 4.

In the residual structure, the upper-level feature map X is directly connected by jumping as the initial result of the partial output, the output can be expressed as H(X) = F(X) + X. When F(X) is 0, it becomes a constant mapping, then H(X) = X is the optimal output. Therefore, the residual structure is designed to converge the residual results to zero, making the model depth increase with better results.

#### 2.3.3. Attentional Mechanisms

Attention Mechanism (AM) is similar to the human eye, which can be understood as a computer vision system simulating the characteristics of the human vision system to focus on the key areas quickly and efficiently, and its core is to select the most critical information for the current task target among many pieces of information. It is widely used in target detection, image segmentation and other computer-vision-related fields. Currently the most commonly used are Channel Attention Mechanism (CAM) and Spatial Attention Mechanism (SAM). CAM combines global average pooling and maximum pooling, compresses the spatial dimension of the feature map, processes the result by the Sigmoid function to obtain the weights of each channel, and multiplies them with the channels of the input feature layer one by one to obtain the feature map.

SAM expands the information on the basis of CAM, and also adopts the global average pooling and maximum pooling fusion, superimposes the result of the channel direction calculation for each feature point, and performs the convolution operation to adjust the number of channels in turn. It then obtains the weight of each feature point by Sigmoid function. Then, we also multiply with the channels of the input feature layer one by one to obtain the feature map and then perform the feature stitching operation.

### 2.4. Principles of the Proposed Algorithm

#### 2.4.1. G-RRDB Terahertz Image-Denoising Model

In this paper, the proposed G-RRDB dense residual network takes 5 RRDB cascade modules as the baseline model and adds 3 Ghost-LKA tandem splicing modules at its opening to extract coarse-grained features of the image. The subsequent denoising network spliced by the cascade of 5 RRDB modules can fuse more image feature information and recover the input image in more detail at the output, while the residual structure is used to connect two DAB modules as branches to the baseline model, which makes the backbone network pay more attention to the features with high weights and discard useless feature information. Then, the features are further extracted by the combined module of two convolutional layers and ReLU activation function, and finally, the features are downscaled by 1 × 1 convolution to obtain the final noise reduction results. The structure of the G-RRDB model is shown in Figure 5a, and Figure 5b shows the structure of the Ghost-LKA module.

#### 2.4.2. Densely Connected Residual Module

In CNN-based image denoising networks, the input of each layer is the output of the previous layer. When the convolutional layers are connected sequentially according to the traditional network connection method, there is a certain loss of information, and as the depth of the network structure increases, the network will suffer from degradation, leading to a significant increase in training error. In contrast, the residual network (ResNet) connects the inputs and outputs directly through a single channel, thus reducing the loss of information.

RRDB is derived from ESRGAN [12], a new type of dense residual connection module, which can be used to alleviate the gradient disappearance problem caused by traditional CNN for image processing, enhance feature propagation and feature reuse, and greatly reduce the number of parameters. The structure of the RRDB module is shown in Figure 6. In the figure, several dense blocks form a dense residual network according to the residual connection structure, and the output of each dense block is multiplied by a weighting factor with the default value of 0.2. In addition, the output and input of each dense block form a residual mapping, and the whole network also forms a residual mapping in the actual operation. By operating in this way, RRDB is able to obtain the global characteristics of the network smoothly. In the figure, the leaky rectified linear unit (LReLU) [20] is the sparse ReLU layer, which is also the activation layer.

RRDB makes full use of the feature information of each convolutional layer in the module by multi-level residual network and dense connection. The original convolutional layers are replaced by optimized residual mapping, and the adjacent convolutional layers are connected by short connections to improve the learning ability of the residual network. Therefore, this paper adopts a dense residual network composed of a cascade of five RRDB modules as the baseline model, and uses its dense connection mechanism to maximize the convolutional information of each layer, thus improving the detailed information of the image, so that the output feature map can contain more feature information of the original image.

#### 2.4.3. Large Kernel Attention Module Based on Ghost Convolutional Structure (Ghost-LKA)

Self-attentive mechanisms [21] are widely used in image processing, but the following problems arise: (1) an image is treated as a one-dimensional sequence, which is not consistent with the two-dimensional architecture of the image; (2) the high complexity of the computation is too resource-intensive for high-resolution images; (3) the traditional sub-attentive module only considers the spatial adaptation, but ignores the channel adaptation of the image. To solve the above problems of the self-attentive mechanism, LKA was created. The LKA module can be decomposed into three convolution blocks, which are depth-wise convolution (DW-Conv) with 5 × 5 convolution kernels, depth-wise dilation convolution (DW-D-Conv) with 7 × 7 convolution kernels and 1 × 1 convolution kernels. The structure of the LKA module is shown in Figure 7.

The module can, in turn, be expressed as:(1)$${\mathrm{Attention}=\mathrm{Conv}}_{1\times 1}\left(\mathrm{DW}-D-\mathrm{Conv}\left(\mathrm{DW}-\mathrm{Conv}\left(F\right)\right)\right)$$
(2)Output=Attention⮿F
where $F\in R^{C\times H\times W}$, and C, H, W denote the length, width and depth of the large kernel convolution, respectively.

While retaining the advantages of convolution and transform, LKA can also solve the problems of poor long-end dependence of convolution and poor adaptability of transform to local information and channel dimensionality. Therefore, in this paper, LKA is used to extract the image features of global perceptual field, and the extracted feature information is more long-end dependent than that of traditional convolution.

Ghost convolution is a lightweight convolutional neural network that can be used as a plug-and-play component. As shown in Figure 8, it first uses a small number of convolutional kernels to extract features from the input feature map, and then further performs a linear change operation on this part of the feature map. Finally, it splices the linearly changed feature map with the non-linearly changed feature map through the residual structure to obtain the output features. This method replaces the conventional convolution by combining a small number of convolution kernels with a lower cost linear change operation, thus effectively reducing the number of parameters and computational effort without affecting the performance of the model.

In summary, this paper combines the structure of Ghost convolution with an LKA module on the basis of the LKA model to form the Ghost-LKA module, as shown in Figure 9. Firstly, the image feature information of the global perceptual field is extracted by the LKA convolution module, then, the feature information of the input image is extracted by the convolution kernel. Afterwards, the feature maps that have undergone the convolution operation and the feature maps that have not undergone the convolution operation are subjected to the stitching operation, and the obtained results are then element-wise summed with the input to obtain the output features.

#### 2.4.4. Improved Attention Mechanism Module DAB

Image denoising algorithms often do not pay enough attention to important image features, resulting in the loss of certain detail information while noise is removed from the image; therefore, adding AM to the denoising model can reduce the loss of image detail information and achieve effective image denoising. The commonly used CAM can learn the features of channel dimension and obtain the channel attention value, but cannot learn the spatial features; while SAM only considers the information of local area and cannot establish the dependence at a long distance. In order to improve the performance of the model to filter image noise and enhance the attention to the target features, this paper proposes an improved attention mechanism module DAB based on CAM and SAM, according to the existing problems.

In the DAB module, the input features first go through the convolution layer for feature extraction and down-sampling to obtain a feature representation of the image, which helps to capture the feature information of the image, and also go through SA and CA operations to obtain the processed feature map s1 and s2. The feature maps are summed to obtain the fused feature maps $X_{a}$. This feature fusion can improve the representation of task-related features by integrating the importance of channels and spaces. The feature maps $X_{a}$ are then processed by local attention and global attention mechanisms, respectively, so as to improve the perception and processing ability of local and global features. The feature maps obtained after the two branches are summed to obtain the final feature maps, and the feature maps are mapped to the range of (0, 1) by the Sigmoid function to obtain the final weight values, which are used as input to process the images by the centralized processing model to obtain the final output. The overall structure of the module is shown in Figure 10. Among them, the specific operations of the centralized processing module are:(3)$${\mathrm{Output}=W\;\times \;s1+X}_{\mathrm{out}}{\;\times \;(1\;\times \;W)\;+\;W\;\times \;s2\;+\;X}_{\mathrm{out}\;}\times \;(1\;-\;W)$$

The model is able to improve the focus on features with higher weights, thus improving the denoising performance.

**Figure 10 foods-12-02819-f010:**
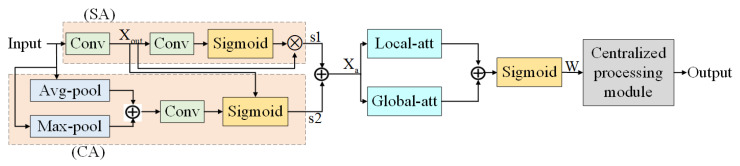
Improved attention mechanism module DAB.

#### 2.4.5. Loss Function

The mean square error (MSE) is a commonly used loss function in tasks such as linear regression to portray the difference between the true and predicted samples. The improved model proposed in this paper uses MSE as the loss function, and the calculation formula is shown as:(4)$$L_{\mathrm{MSE}}=-\frac{1}{N}\sum_{i-1}^{N}y_{i}-{\hat{y}}_{i}$$
where N indicates the sample size, $y_{i}$ denotes the label of the ith sample, and ${\hat{y}}_{i}$ denotes the predicted probability of the ith sample. The smaller the value of $L_{\mathrm{MSE}}$, the smaller the difference between the predicted and true values of the model, which means the model performance is better. MSE can be seen as the average of the model’s squared error on the predicted values and is therefore more sensitive to outliers. When there are outliers in the sample, MSE may be affected resulting in reduced model performance.

### 2.5. Evaluation Indicators

Evaluation metrics can objectively evaluate the effect of various algorithms and accurately measure the algorithm performance. In this paper, we adopt the commonly used image quality evaluation metrics PSNR and SSIM to quantitatively evaluate and analyze the denoised images.

PSNR is mainly used for quality evaluation of the reconstructed signal and represents the ratio of the maximum power achieved by the signal to the destructive noise power that affects the accuracy of the reconstructed signal. The specific expressions are:(5)$$\mathrm{PSNR}=10\;\times \;\mathrm{lg}(\frac{{\mathrm{MAX}}_{1}^{2}}{\mathrm{MSE}})=20\;\times \;\mathrm{lg}(\frac{{\mathrm{MAX}}_{1}}{\sqrt{\mathrm{MSE}}})$$
where MAX usually refers to the image grayscale level, MSE means mean squared error.

SSIM is a metric used to measure the similarity of two digital images. The similarity of two images is evaluated in terms of brightness, contrast and structure, and its value ranges from 0 to 1. A larger value indicates better image quality. The specific expressions are:(6)$$\mathrm{SSIM}(x,y)=\frac{{(2\mu }_{x}\mu_{y}{+c}_{1}{)(2\sigma }_{\mathrm{xy}}{+c}_{2})}{{(\mu }_{x}^{2}{+\mu }_{y}^{2}{+c}_{1}{)(\sigma }_{x}^{2}{+\sigma }_{y}^{2}{+c}_{2})}$$
where $\mu_{x}$ means the average value of x, $\mu_{y}$ is the average value of y, $\sigma_{x}^{2}$ is the variance of x, $\sigma_{y}^{2}$ is the variance of y, and $\sigma_{\mathrm{xy}}$ represents the covariance of x and y. $c_{1}{=(k}_{1}{L)}^{2}$, $c_{2}{=(k}_{2}{L)}^{2}$, of which $k_{1}=0.01$, $k_{2}=0.03$, and L indicates the dynamic range of the pixel value.

## 3. Results and Discussion

### 3.1. Model Training

In this experiment, the data set consists of three groups of THz spectral images of normal wheat grains, three groups of slightly moldy wheat, three groups of moderately moldy wheat and three groups of seriously moldy wheat in different frequency domains, and the training and testing sets are divided in a ratio of 9:1. In this paper, the hardware configuration for model training and validation is: Intel Core i7-11700 (2.5 GHz) with 64 GB memory and NVIDIA RTX 3090 GPU model; software environment is: python3.7, pytorch2.0, and CUDA framework with version 11.3 is used for accelerated computing. To ensure the objectivity and realism of the model performance comparison, the Adam optimizer was used uniformly in the experiment, the initial learning rate was set to 1 × 10^−3^, the batch size was set to 8, and the epoch was set to 70. Since the internal protein and other compounds of wheat in different mold periods will change, the internal image features will also change accordingly. The background of the original wheat THz image was removed and the THz images of wheat after background removal are shown in Figure 11. Then, the training samples and their corresponding labels were used as the input of the network for image denoising.

During the process of mold growth, hydrolytic enzymes secreted by microorganisms hydrolyze the proteins and carbohydrates inside the wheat grain, so that the nutrients it contains are reduced and more mold is produced, which gradually contaminates and encapsulates the wheat grain. Therefore, the gradual change of the color area inside the image indicates the reduction of nutrients, as shown in Figure 11.

### 3.2. Experimental Results and Discussion

The THz images after removing the background still suffer from the problem of unremarkable features and poor image quality, which can affect the subsequent detection accuracy of moldy wheat and thus lead to unnecessary grain loss. To verify the denoising performance of the network proposed in this paper, this study uses block matching and 3D filtering (BM3D) [22], ADNet [23], DnCNN, CBDNet, the baseline model and the combined network of baseline model with DAB to compare with the denoising network proposed in this paper, and the denoising effects of the above seven denoising models are shown in Table 1. The images shown are all denoised 2D images corresponding to the 50th frequency domain point of the spectrum. The BM3D model leads to the loss of image edges, the DnCNN loses some image details, the baseline model and the combined network of the baseline model and DAB module have good denoising effects, but the image features are still not prominent enough. The results show that the proposed model can effectively remove noise while preserving the edge information, and the denoising effect is better.

As shown in Figure 12, the detailed features of the image processed by G-RRDB are highlighted and the image quality is effectively improved.

The comparison of the PSNR and SSIM evaluation results of the above seven denoising models is shown in Table 2. The PSNR/SSIM values in Table 2 were obtained by averaging the values of all the images in the test set for each type of sample. From the comparison in the table, it can be seen that the PSNR and SSIM values of the models in this paper are the highest level compared with the traditional models. Taking slightly moldy wheat as an example, its PSNR and SSIM values are 0.41 dB and 0.01 higher than those of the baseline model, respectively, and the PSNR value is 0.02 dB higher than that of the combined network of the baseline model with DAB module, indicating that the added DAB module and Ghost-LKA module can effectively improve the denoising performance of the baseline model. Additionally, they are higher than those of the BM3D model by 3.20 dB and 0.05, respectively, indicating that the denoising effect is greatly improved.

### 3.3. Model Validation

To verify the effect of network denoising performance under different levels of noise, this paper adds four degrees of Gaussian noise (20 dB, 30 dB, 40 dB, 50 dB) for the proposed model, thus restoring the image acquisition environment with different degrees of noise that may exist. Taking the seriously moldy wheat samples as an example, the denoising results are shown in Table 3. The PSNR/SSIM values in Table 3 were obtained by averaging the values of all the images in the test set of seriously moldy wheat, and these images were obtained after four levels of Gaussian noise addition.

As can be seen in Table 3, the G-RRDB model achieves optimal values for image denoising in all four degrees of Gaussian noise environments. Under 20 dB of noise, compared to the relatively newest ADNet, its PSNR and SSIM are improved by 1.53 dB and 0.03, respectively. Compared with the traditional model DnCNN, the PSNR and SSIM are improved by 1.69 dB and 0.02, respectively, which shows that the best denoising effect of the model proposed in this paper is still achieved in the noise-added environment.

To further verify the effectiveness of thee G-RRDB model for THz image denoising, this paper used VGG16 [24] network to classify four kinds of samples for verification, and the experimental data were divided into denoised images of baseline model, denoised images of the combined baseline model and DAB module, and denoised images of G-RRDB. The classification results are shown in Table 4. For the obtained data of the classification experiments, the prediction result of each model for each kind of sample was obtained by averaging the results of two replicate experiments.

The effectiveness of the G-RRDB THz image-denoising model was validated using VGG16 as shown in Table 4. Taking normal wheat samples as an example, the prediction accuracy was improved from 91.1% and 92.6% to 92.8%, compared with the classification results of the denoised images of the baseline model and that of the combined baseline model and DAB module, which were improved by 1.7% and 0.2%, respectively. The results show that VGG16 has good prediction results for the denoised images of G-RRDB.

### 3.4. Discussion

In this paper, a G-RRDB THz image-denoising network is compared with 6 image-denoising models, including BM3D, ADNet, CBDNet, DnCNN and the baseline model, etc. The experimental results show that G-RRDB provides images with the maximum PSNR and SSIM values, and has the best denoising effect among these seven models. Moreover, after the verification of four sets of Gaussian noise, the PSNR and SSIM values of the images after G-RRDB denoising reach 35.86 dB and 0.98, respectively, and the denoising effect is still the best. Furthermore, compared to the denoised images obtained from the baseline model and the combined baseline model with the DAB module, the classification accuracy of the image processed by G-RRDB is the highest. The results of the above experiments indicate that the G-RRDB THz image-denoising model proposed in this paper effectively reduces THz image noise, improves image quality, highlights image features and has excellent results.

In the proposed network, the incorporation of Ghost-LKA and DAB modules inevitably increases the number of parameters compared to the baseline model. Therefore, future improvements should focus on achieving lighter weight and higher precision, thus providing greater image denoising capability. Meanwhile, in the future, we will expand the variety of samples and apply this algorithm to other agricultural quality tests so as to improve its generalizability. Finally, our next step is to apply the algorithm to the early detection of moldy wheat and provide an early warning to nip the wheat in the bud where mold is occurring by controlling the temperature to 12° or less and ventilating the wheat in time, thereby safeguarding the quality of the wheat and reducing post-production grain loss.

## 4. Conclusions

To address the issues of low contrast and lack of edge information in the original THz image, this paper proposes a THz image-denoising model called G-RRDB to improve image quality. The method first incorporates the LKA convolutional block into the Ghost convolutional model structure, thus proposing the Ghost-LKA module. This enhancement allows the Ghost convolutional module to acquire the global perceptual field while extracting feature information, thereby addressing the problem of incomplete feature extraction and improving the detection accuracy. Additionally, the Ghost-LKA module is connected at the beginning of the baseline model. Furthermore, this paper proposes an improved attention mechanism module, DAB, which maximizes the retention of important image features through feature fusion. The DAB module is connected to the baseline model through the residual structure, preventing gradient disappearance and accelerating the convergence speed of the model. In comparison to traditional algorithms such as BM3D and DnCNN, the proposed model exhibits a better denoising effect, with PSNR and SSIM values reaching 35.86 dB and 0.98, respectively. By adding different levels of Gaussian noise to the moldy wheat images, the PSNR value of the G-RRDB denoised images reaches 35.86 dB and the SSIM value reaches 0.98, and the denoising effect was still better than other comparison models. Additionally, the classification accuracy of G-RRDB denoised images of wheat with four mold degrees is verified using VGG16, and the classification accuracy reaches 92.8%, which represents a 1.7% improvement compared to the classification results of baseline-model-denoised images. Thus, this study presents a new approach for enhancing the quality of THz images and improving the accuracy of agricultural product quality detection.

## Figures and Tables

**Figure 1 foods-12-02819-f001:**
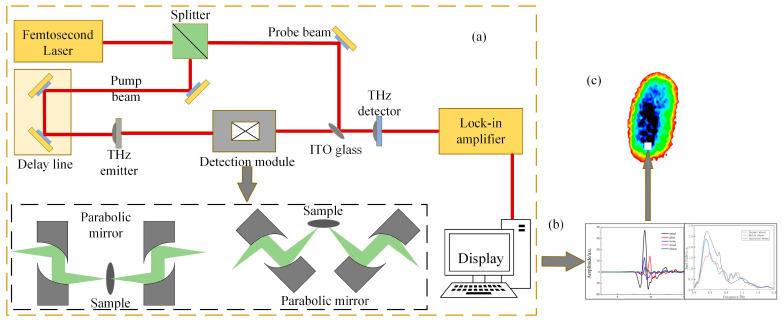
Schematic diagram of THz 3D chromatography imaging system. (**a**) Basic optical path structure of the system, (**b**) THz spectrum of the sample, (**c**) THz sample of the sample.

**Figure 2 foods-12-02819-f002:**
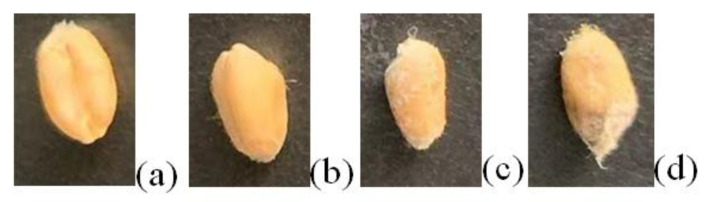
Wheat samples with different degrees of mold: (**a**) normal; (**b**) slightly moldy; (**c**) moderately moldy; (**d**) seriously moldy.

**Figure 3 foods-12-02819-f003:**
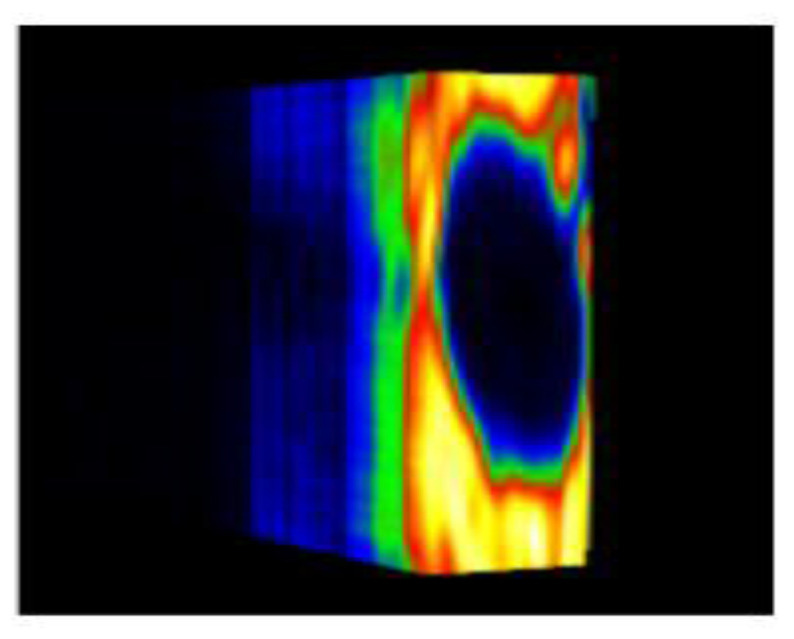
The 80th layer Thz image of slightly moldy wheat.

**Figure 4 foods-12-02819-f004:**
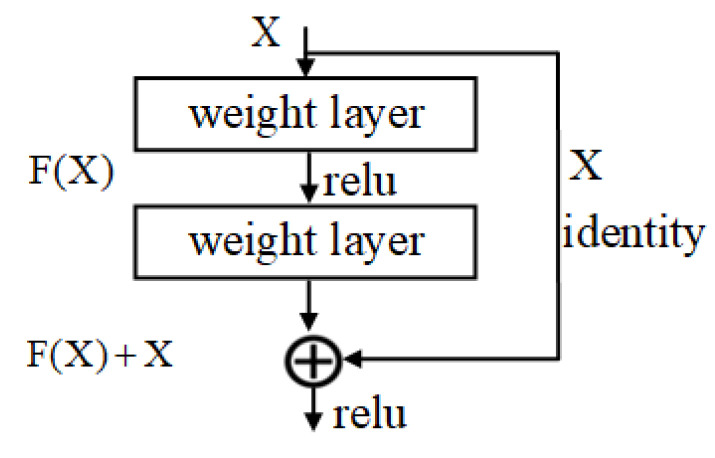
Schematic diagram of the residual structure.

**Figure 5 foods-12-02819-f005:**
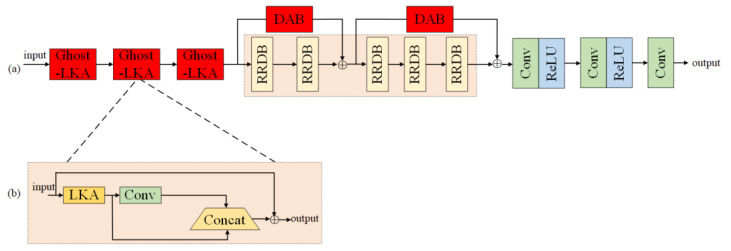
G-RRDB terahertz spectral image enhancement model. (**a**) The overall structure of G-RRDB, (**b**) Structure of Ghost-LKA.

**Figure 6 foods-12-02819-f006:**
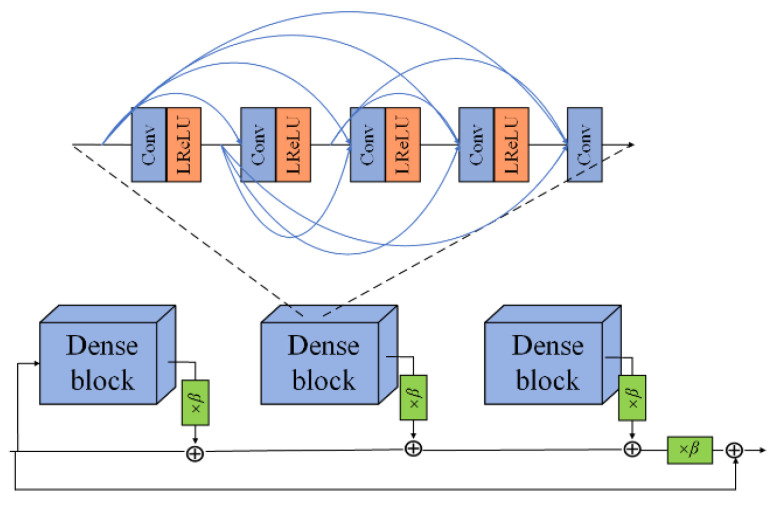
RRDB module.

**Figure 7 foods-12-02819-f007:**
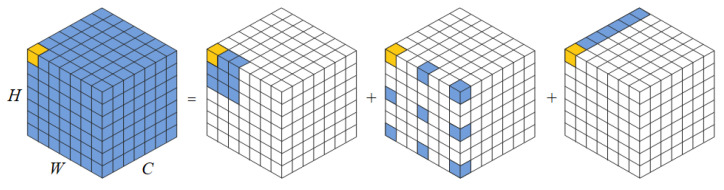
Composition of the LKA module.

**Figure 8 foods-12-02819-f008:**
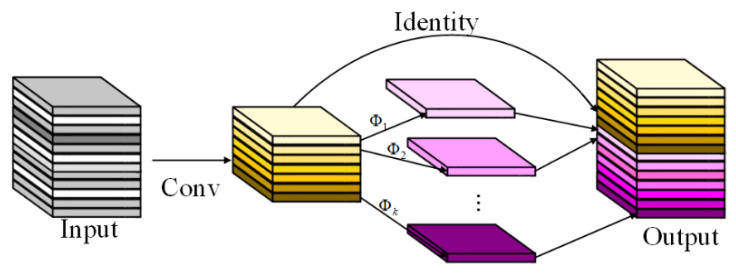
Ghost convolution module.

**Figure 9 foods-12-02819-f009:**
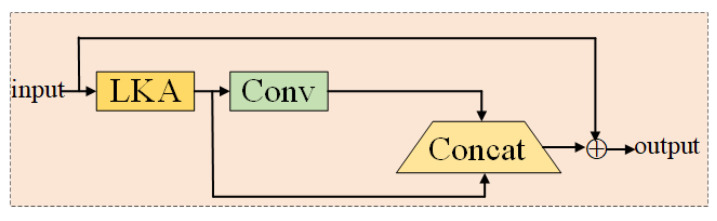
Ghost-LKA module.

**Figure 11 foods-12-02819-f011:**
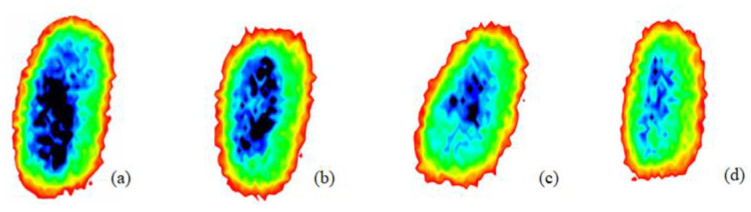
Terahertz image with background removed: (**a**) normal; (**b**) slightly moldy; (**c**) moderately moldy; (**d**) seriously moldy.

**Figure 12 foods-12-02819-f012:**
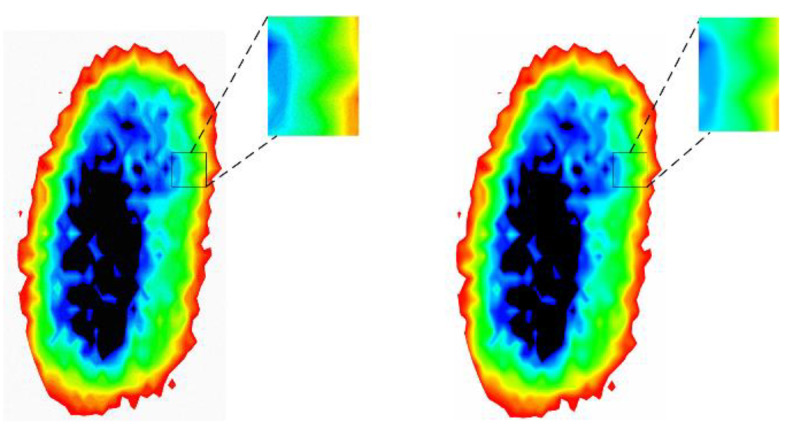
Comparison of original normal wheat image and G-RRGB denoised image.

**Table 1 foods-12-02819-t001:** Comparison of the effects of seven denoising networks.

Algorithm	BM3D	ADNet	DnCNN	CBDNet	Baseline	Baseline + DAB	G-RRDB
Species
Normal	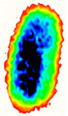	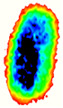	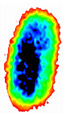	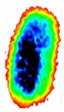	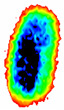	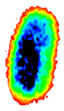	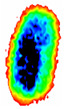
Slightly moldy	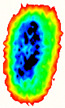	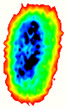	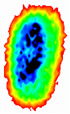	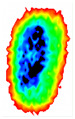	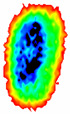	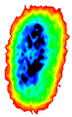	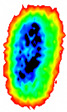
Moderately moldy	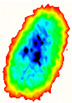	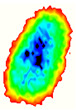	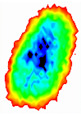	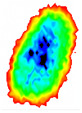	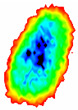	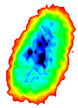	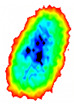
Seriously moldy	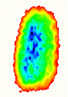	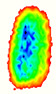	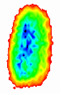	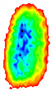	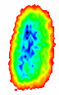	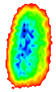	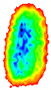

**Table 2 foods-12-02819-t002:** Comparison of PSNR/SSIM results based on seven denoising networks.

Species	Normal		Slightly Moldy		Moderately Moldy		Seriously Moldy	
Algorithm	PSNR/dB	SSIM	PSNR/dB	SSIM	PSNR/dB	SSIM	PSNR/dB	SSIM
BM3D	32.94	0.94	32.78	0.93	32.46	0.93	32.92	0.94
ADNet	34.32	0.95	34.80	0.96	35.04	0.96	34.77	0.96
CBDNet	34.41	0.95	35.05	0.96	35.78	0.97	35.11	0.96
DnCNN	33.57	0.96	34.19	0.96	34.79	0.97	34.22	0.96
Baseline	34.86	0.96	35.57	0.97	36.13	0.98	35.63	0.97
Baseline + DAB	35.23	0.97	35.96	0.98	36.59	0.98	36.02	0.98
G-RRDB	35.32	0.97	35.98	0.98	36.62	0.98	36.21	0.98

**Table 3 foods-12-02819-t003:** Denoising results with different levels of Gaussian noise.

Images	Model	20 dB		30 dB		40 dB		50 dB	
		PSNR/dB	SSIM	PSNR/dB	SSIM	PSNR/dB	SSIM	PSNR/dB	SSIM
Seriously moldy	BM3D	31.43	0.93	31.24	0.93	30.95	0.92	30.55	0.90
	ADNet	34.33	0.95	33.76	0.94	33.07	0.91	32.22	0.94
	CBDNet	34.40	0.96	34.28	0.96	34.01	0.96	33.82	0.95
	DnCNN	34.17	0.96	34.02	0.96	33.78	0.96	33.44	0.95
	Baseline	35.49	0.97	35.26	0.97	34.96	0.97	34.60	0.96
	Baseline + DAB	35.85	0.97	35.59	0.97	35.23	0.97	34.80	0.97
	G-RRDB	35.86	0.98	35.60	0.98	35.26	0.97	35.17	0.97

**Table 4 foods-12-02819-t004:** Comparison of classification results.

	Prediction Results (%)		
Images	Baseline	Baseline + DAB	G-RRDB
Normal	91.1	92.6	92.8
Slightly moldy	90.2	91.0	91.6
Moderately moldy	90.3	92.2	92.5
Seriously moldy	90.2	91.8	92.2

## Data Availability

The data is not publicly available due to confidentiality restrictions. Please contact the corresponding author for questions related to data availability.

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
