# Peer review of "G-RRDB: An Effective THz Image-Denoising Model for Moldy Wheat"

_foods, 2023, doi:10.3390/foods12152819_

Round 1

Reviewer 1 Report

Dear Authors,

The subject of the study is interesting and topical, with scientific and practical importance.

The introduction is presented correctly, in accordance with the subject. Numerous scientific articles, in concordance to the topic of the study, were consulted.

Methodology of the study was clearly presented, and appropriate to the proposed objectives.

The obtained results are important and have been analyzed and interpreted correctly, in accordance with the current methodology.

The discussions are appropriate, in the context of the results, and was conducted compared to other studies in the field.

The scientific literature, to which the reporting was made, is recent and representative in the field.

Some suggestions and corrections were made in the article.

The following aspects are brought to the attention of the authors.

1.

Space between words and parentheses with the bibliographic sources cited in the text

e.g.

page 1, row 36

“vision [1] and” instead of “vision[1]and”

Please check the entire article and make corrections as necessary.

2.

Please check the settings for chapter titles, subchapters, and for figure titles

3.

Page 10, rows 366 – 368

„Since the internal protein and other compounds of wheat in different mold periods will change, the internal image features will also change accordingly.”

It would be interesting if the reflection of the grains quality indices in the images could be presented.

Which grains quality index was reflected more strongly in the images?

4.

Conclusions chapter

The Conclusions chapter is missing.

It would be appropriate to present a Conclusions chapter.

5.

References

According to Instructions for Authors and Microsoft Word template, Foods journal,

Author 1, A.B.; Author 2, C.D. Title of the article. Abbreviated Journal Name Year, Volume, page range.

Include the digital object identifier (DOI) for all references where available.

e.g.

page 13, rows 452 – 453

J. Agric. Food Res.” instead of “Journal of Agriculture and Food Research

Abbreviated Journal Name

Please check and correct, if necessary.

Reviewer 2 Report

1. It is necessary to present the results of statistical processing of the obtained results of experimental studies.

2. It is necessary to present the variety and varietal characteristics of wheat grain during experimental studies.

3. In the text of the manuscript, it is necessary to indicate the general view of the experimental setup for the study of terahertz chromatography.

4. It is necessary to indicate the degree of damage to wheat grains by the volume of grains, the graduation of damage to grains used in the text of the manuscript is not entirely clear: normal grains, moldy grains ...

5. In addition, it is necessary to present a method for diagnosing substandard wheat grain, as well as a method for its infection.
